# Promising Antibiofilm Agents: Recent Breakthrough against Biofilm Producing Methicillin-Resistant *Staphylococcus aureus*

**DOI:** 10.3390/antibiotics9100667

**Published:** 2020-10-03

**Authors:** Marwa I. Abd El-Hamid, El-sayed Y. El-Naenaeey, Toka M kandeel, Wael A. H. Hegazy, Rasha A. Mosbah, Majed S. Nassar, Muhammed A. Bakhrebah, Wesam H. Abdulaal, Nabil A. Alhakamy, Mahmoud M. Bendary

**Affiliations:** 1Department of Microbiology, Faculty of Veterinary Medicine, Zagazig University, Zagazig 44511, Egypt; mero_micro2006@yahoo.com (M.I.A.E.-H.); sayedmyn@hotmail.com (E.-s.Y.E.-N.); 2Specialist of Laboratory Medical Analysis, Almokhtabar Private Laboratories, Zagazig 44511, Egypt; ninejuly88@yahoo.com; 3Department of Microbiology and Immunology, Faculty of Pharmacy, Zagazig University, Zagazig 44511, Egypt; waelmhegazy@daad-alumni.de; 4Fellow Pharmacist, Infection Control Unit, Zagazig University Hospital, Zagazig 44511, Egypt; rashamosbah@hotmail.com; 5Life Science and Environment Research Institute, King Abdulaziz City for Science and Technology (KACST), P.O. Box 6086, Riyadh 11442, Saudi Arabia; mnassar@kacst.edu.sa (M.S.N.); Mbakhrbh@kacst.edu.sa (M.A.B.); 6Department of Biochemistry, Cancer Metabolism and Epigenetic Unit, Faculty of Science, King Abdulaziz University, Jeddah 21589, Saudi Arabia; whabdulaal@kau.edu.sa; 7Department of Pharmaceutics, Faculty of Pharmacy, King Abdulaziz University, Jeddah 21589, Saudi Arabia; nalhakamy@kau.edu.sa; 8Department of Microbiology and Immunology, Faculty of Pharmacy, Port Said University, Port Said 42511, Egypt

**Keywords:** MRSA, antibiofilm, antimicrobial, ZnO-NPs, proteinase-K, HAM

## Abstract

Multidrug resistant (MDR) methicillin-resistant *Staphylococcus aureus* (MRSA) is a superbug pathogen that causes serious diseases. One of the main reasons for the lack of the effectiveness of antibiotic therapy against infections caused by this resistant pathogen is the recalcitrant nature of MRSA biofilms, which results in an increasingly serious situation worldwide. Consequently, the development of innovative biofilm inhibitors is urgently needed to control the biofilm formation by this pathogen. In this work, we thus sought to evaluate the biofilm inhibiting ability of some promising antibiofilm agents such as zinc oxide nanoparticles (Zno NPs), proteinase K, and hamamelitannin (HAM) in managing the MRSA biofilms. Different phenotypic and genotypic methods were used to identify the biofilm producing MDR MRSA isolates and the antibiofilm/antimicrobial activities of the used promising agents. Our study demonstrated strong antibiofilm activities of ZnO NPs, proteinase K, and HAM against MRSA biofilms along with their transcriptional modulation of biofilm (intercellular adhesion A, *ica*A) and quorum sensing (QS) (*agr*) genes. Interestingly, only ZnO NPs showed a powerful antimicrobial activity against this pathogen. Collectively, we observed overall positive correlations between the biofilm production and the antimicrobial resistance/*agr* genotypes II and IV. Meanwhile, there was no significant correlation between the toxin genes and the biofilm production. The ZnO NPs were recommended to be used alone as potent antimicrobial and antibiofilm agents against MDR MRSA and their biofilm-associated diseases. On the other hand, proteinase-K and HAM can be co-administrated with other antimicrobial agents to manage such types of infections.

## 1. Introduction

*Staphylococcus aureus* (*S. aureus*) is a widely distributed pathogen responsible for a variety of disease conditions. It can cause invasive infections in humans and animals with high mortality rates worldwide [1]. Adding to this, the rapid emergence and dissemination of multi-drug resistant (MDR) strains, especially methicillin resistant *S. aureus* (MRSA), has become an obstacle to public health.

Numerous different diseases are associated with *S. aureus* infection. This has been attributed to the expression of a large repertoire of virulence genes such as coagulase (*coa*), *S. aureus* protein A (*spa*), Panton–Valentine leucocidin (*pvl*), toxic shock syndrome toxin (*tst*), exfoliative toxins A (*eta*) and B (*etb*), and staphylococcal enterotoxins A–G (*sea*–*seg*), which cause harmful toxic effects to the host [2,3]. The staphylococcal virulence factors are under the control of the accessory gene regulator (*agr*) quorum sensing (QS) system. The *agr* system increases the production of many virulence factors and decreases the expression of several colonization factors under conditions of high cell density.

Like many other organisms, a key aspect of the recalcitrant nature of *S. aureus* infections is related to their adherence and invasion, which is associated with their inherent ability to form biofilms on natural and abiotic surfaces [4]. Indeed, staphylococcal biofilm associated lethal infections are particularly more difficult to eradicate and its treatment has become extremely challenging and costly. Such biofilms protect the cells not only from the host immune defenses, but also from the effects of chemotherapeutic agents [5]. In this way, *S. aureus* displays increasing resistance to multiple antimicrobials within their biofilms [6]. This gave rise to a great concern, because it can cause an enormous burden to the healthcare systems [7].

Biofilms are complex and organized communities of sessile microorganisms held together by a self-synthesized organic matrix of extracellular polymeric substances, thereby exhibiting an altered phenotype compared with the free planktonic counterparts [8]. The biofilm formation occurs via bacterial adherence to a substratum, which is facilitated by the expression of various proteins that bind to one or more host extracellular matrix factors. They are collectively called microbial surface components, which recognize adhesive matrix molecules (MSCRAMMs) [9]. In the bacterial accumulation phase, the development of an actual biofilm is mainly mediated by the polysaccharide intercellular adhesin (PIA), whose production and excretion is directed by the *ica*ADBC operon-coded enzymes. The QS system has also been shown to regulate biofilm formation and dispersal in *S. aureus* [10]. This process is mediated by the production of small signaling molecules as a universal language for cell-to-cell communication [11]. One of the most intriguing is the apparent role of QS as a multicellular strategy in the resistance of biofilm to antimicrobials relying on the exchange of chemical signals between cells [12].

A disturbing trend in *S. aureus* is if the MDR strains, especially MRSA, are capable of forming the biofilm; the biofilm inherent antimicrobial resistance coupled with the ideal protection granted by this barrier may contribute to the recalcitrance to the antibiotic therapy [13]. Hence, the conjunction of all the above mentioned factors together drives the need for developing new alternative strategies to effectively tackle MDR and biofilm associated infections. In recent years, many synthetic and natural antibiofilm compounds have been extensively employed in an attempt to control this complex mode of bacterial life in clinical settings [14], but limited information is available regarding this mechanism on MRSA biofilm in the food industry [15].

In fact, many studies in the last decades have determined the possibility of preventing *S. aureus* biofilm formation. However, biofilm mediated persistent infections shifted the focus on discovering innovative biofilm dispersal approaches [16]. Apart from this, the goal of identifying new therapeutic targets is more than welcome. Interfering with *S. aureus agr* QS signaling system using QS inhibitors (QSIs) can provide a promising antibiofilm option to combat biofilm related infections. Al-Shabib and his collaborators stated that scarce data are already available on these aspects, especially in the Arab world [15].

The problem of biofilm-associated infections could be combated by several compounds targeting biofilm matrices of MRSA. Among the promising compounds, hamamelitannin (HAM), a naturally occurring polyphenolic relatively inexpensive compound extracted from the bark of *Hamamelis virginiana* and belonging to the family of tannins, has been shown to display a wide range of antimicrobial activity. It seems to prevent MRSA biofilm-associated infections via blocking the QS system, thereby reducing its virulence [17]. Other methods like the treatment of *S. aureus* biofilms with proteases such as proteinase K could be used for controlling the biofilm formation. It has frequently been used as an efficient biofilm removal agent that leads to *S. aureus* biofilm disassembly, presumably through degradation of the surface structures [18]. Alternatives to these methods, the recent evolution in nanotechnology has been confirmed to be an effective approach used to control the biofilm formation [14]. Among the nano-sized metal oxides, zinc oxide (ZnO) has gained much attention thanks to its distinctive physical and chemical properties such as its high surface-to-volume ratio, long-term stability, and low cost [19]. A recent finding supported the notion that ZnO NPs possessed efficient antibiofilm activity against *S. aureus* [20].

Considering the above mentioned aspects, we endeavored to determine the biofilm forming capacity and the antibiotic resistance patterns of MRSA isolates, and simultaneously assess the correlation between them to comprehend the role of MRSA biofilms in the resistance. Most importantly, this study has described the biofilm development and dispersal mechanisms. To address this idea, we selected several compounds targeting biofilm matrices of MRSA isolated from mastitis milk and meat samples along with human clinical specimens. In this context, we sought to evaluate the antibiofilm potentials of ZnO NPs, proteinase K, and HAM and, simultaneously, we endeavored to ascertain whether these approaches could effectively interfere with QS systems of the examined MRSA isolates.

## 2. Results

### 2.1. Characterization of MRSA Biofilm Producers

Out of 500 human, animal, and food samples, 222 MRSA isolates were identified using different phenotypic and genotypic characterization methods. About 27% (*n* = 60) of these isolates were identified phenotypically as biofilm producers depending on growth onto Congo red agar and adherence to microtiter plate (MTP). Surprisingly, the highest percentage of biofilm producing MRSA was among the strains isolated from human samples (32 isolates out of 200 samples, 16%), followed by food (11 out of 100, 11%) and animal (17 out of 200, 8.5%) ones. Among the human samples, wound swabs possessed the highest percentages of biofilm forming MRSA (20.3%). On the other hand, the lowest percentages were detected among sputum and cerebrospinal fluid (CSF) samples (13.2 and 9.1%, respectively).

### 2.2. Antibiogram of Biofilm Producing and Non-Producing MRSA

Concerning the antibiogram results, all isolates were resistant to methicillin and cefoxitin (100% each), confirming that they were all MRSA. All biofilm producing MRSA isolates showed high percentages of resistance against all the tested antimicrobial agents in opposition to the non-biofilm producer ones, confirming the association between antimicrobial resistance and biofilm production (Table 1). Antimicrobial susceptibility patterns showed that the maximum resistance of the non-biofilm producing isolates was against ciprofloxacin (48.8%). Meanwhile, a relatively low resistance rate of these isolates was observed against imipenem (3.7%) (Table 1). Regarding the biofilm producing MRSA, all these isolates were resistant to three or more different categories of the investigated antimicrobials, suggesting them to be MDR. Depending on their sources, chloramphenicol and imipenem had no antimicrobial activities against food isolates, whereas animal isolates exhibited higher resistance rates against chloramphenicol and erythromycin (94.1 and 88.2% respectively). Twenty human isolates (62.5%) were uniformly resistant to ciprofloxacin, trimethoprim-sulfamethoxazole, and gentamicin (Figure 1 and Table 1). Despite the existence of large number of vancomycin resistant isolates, vancomycin is still the most effective drug against both human and animal isolates in comparison with other antimicrobials. Meanwhile, the food isolates showed a relative sensitivity to rifamycin SV (45.5%) (Figure 1).

### 2.3. Genetic Characterization of MRSA Isolates

In parallel to our phenotypic methicillin resistance results, all examined isolates harbored *mec*A, confirming that they were all MRSA isolates. Moreover, all biofilm producing isolates (*n* = 60) possessed intercellular adhesion A (*icaA*) gene.

Concerning the toxin gene profiles, biofilm producing MRSA isolates were more toxigenic than the non-biofilm producing ones, except for the isolates those harboring the *eta* toxin gene (Table 1). Analyzing the toxin gene profiles of the biofilm producing MRSA isolates (*n* = 60) revealed that the most prevalent toxin gene was *pvl* (53.3%), whereas *eta* was the least detected one among those isolates (6.7%) (Table 1).

Regarding the *agr* genotyping, *agr* I was the common genotype among food, animal, and human biofilm producing and non-producing MRSA isolates. In total, *agr* I was observed in 77.8 and 65% of our non-biofilm and biofilm MRSA producers, respectively. On the other hand, *agr* II was the least recovered genotype among biofilm producing and non-producing MRSA isolates (5 and 3.7%, respectively) (Table 1).

### 2.4. Correlation Between Biofilm Production and Antimicrobial Resistance, Toxin Genes, and agr Genotypes

Correlation analyses used to unravel the extent of association between biofilm production ability and antimicrobial resistances, toxin genes, and *agr* genotypes revealed that there was no significant correlation between biofilm production and toxin genes. Except for ciprofloxacin and gentamicin, there was a positive correlation between biofilm production and the resistance to other antimicrobials. Only amoxicillin-clavulanic acid and ceftriaxone resistances showed a non-significant positive correlation. The highest positive correlation with biofilm production was observed to vancomycin and imipenem resistances (*R*-value = 0.86 and 0.64, respectively, *p* < 0.05). Considering *agr* genotypes, *agr* types II and IV correlated positively in a significant manner opposite to *agr* I, which correlated negatively with the biofilm production (Figure 2).

### 2.5. Phenotypic Analysis of Antibiofilm Activities of Zinc Oxide Nanoparticles (ZnO NPs), Proteinase K, and Hamamelitannin (HAM)

By analyzing the results of quantitative phenotypic detection of biofilm post exposure to the tested materials using the MTP method, the capacity of all biofilm producing MRSA isolates were highly decreased post exposure to all examined substances, with efficacy rates ranging from 99.06 to 99.95%. Therefore, all the tested materials proved pronounced antibiofilm effects on the examined isolates, especially when compared with the untreated ones.

Overall, the promising antibiofilm agents had powerful antibiofilm capacity with the following order: ZnO NPs, HAM, and proteinase K, with mean inhibitory capacity percentages of 99.73%, 99.66%, and 99.41%, respectively. The activity order for the promising agents was identical against human isolates (Table 2). Of note, there were no significant differences between the effects of these antibiofilm substances on the biofilm capacity of human, food, and animal isolates (*p* > 0.05).

### 2.6. Biofilm Associated Genes Expressions Post Exposure to the Antibiofilm Agents

Analysis of the quantitative reverse transcription-polymerase chain reaction (qRT-PCR) results demonstrated that the transcriptional levels of *ica*A and *agr* genes were remarkably decreased with mean values of fold changes up to 0.15- and 0.08, respectively) after using ZnO NPs in prevention of MRSA biofilm production (Figure 3). Likewise, the use of proteinase K and HAM also caused the downregulation of both genes. In proteinase K, the reduction in the expression of *ica*A and *agr* genes was in equal rates with mean values of fold changes up to 0.55- and 0.59, respectively). Similarly, HAM exhibited reduced expression levels of *ica*A and *agr* genes with mean values of fold changes up to 0.32- and 0.22, respectively (Table 2 and Figure 3). The repression of these genes under ZnO NPs, proteinase K, and HAM treatment confirmed the observed reduction in biofilm formation of MRSA. It was observed that ZnO NPs has the highest effect on the down regulation of both *ica*A and *agr* genes, followed by HAM and proteinase K (Figure 3). This collaborated with the pervious phenotypic results using the MTP method. Of note, there was a strong positive correlation between the expression of both *agr* and *icaA* genes for all types of the investigated isolates (*R* = 1). There were significant differences between the effects of all examined antibiofilm materials on the down regulation of *agr* I gene (*p* < 0.05), but there were no significant variations between the effects of all antibiofilm substances (*p* > 0.05), except HAM (*p* < 0.05), on the down regulation of *ica*A gene. For all human, animal, and food isolates, the downregulation of *agr* gene expression was found to be the most pronounced effect of the three promising antibiofilm agents (Figure 3).

### 2.7. Antimicrobial Activities of ZnO NPs, Proteinase K, and HAM

The antimicrobial activity of ZnO NPs on all tested isolates was approved using the agar well diffusion method. The diameters of inhibition zones around each well with ZnO NPs solution at 100 μg/mL ranged from 12 to 14 mm. These results demonstrated that ZnO nanoparticles were effective to inhibit the growth of biofilm producing MRSA even at low concentrations. On the other hand, no antimicrobial activities were observed to both proteinase K and HAM.

## 3. Discussion

In recent years, the increasing incidence of diseases caused by biofilm-associated organisms has been noted globally. Biofilms pose a serious problem for public health, because biofilm-producing microorganisms exhibit dramatically increased resistance to both antimicrobial agents and host immune response. Of note, the increase in the incidence of MDR bacterial and fungal strains makes many public crises [21] as the conventional antibiotic medications are inadequate at eradicating these pathogens.

In the current research, all biofilm producing MRSA were MDR and showed insusceptibility to methicillin and cefoxitin. This finding is parallel with a previous report in Nepal, where 86.7% of biofilm-positive *S. aureus* isolates were MDR and 43.3% of biofilm producers were MRSA, whereas no MDR or MRSA isolates were noted among any of the biofilm nonproducers [22]. These results confirm that the biofilm formation may be one of the pivotal factors for increasing the resistance towards the commonly used antimicrobials. The mechanism of MDR in the biofilm-forming strains may be attributed to the close cell contact in the biofilm, which facilitates the easy transfer of plasmids containing MDR genes among them.

In the clinical treatment, biofilm formation and drug-resistance make bacterial eradication more difficult. Therefore, the correlation between the biofilm production and the drug resistance should be analyzed. As was previously announced [23], our results confirmed that the biofilms play an important role in increasing the antibiotic resistance, and there was a positive correlation between antimicrobial resistance and biofilm formation. The optimization of the usage of vancomycin and imipenem is a promising strategy to reduce the treatment failures against biofilm-associated infections [24]. Unfortunately, there was a strong positive correlation between vancomycin/imipenem resistances and biofilm production among our isolates, which provided a negative impact on the management of MRSA infections. Several studies reported the low activity of vancomycin against biofilm-embedded *S. aureus*. Previously, it was noted that there was no statistically significant differences between the *S. aureus* remaining in a non-treated biofilm or in a biofilm exposed to vancomycin for 24 h [25]. In this context, it was documented that there was an increase in the percentages of the *S. aureus* resistance in spite of the usage of a very high constant concentration of vancomycin (5000 mg/L) for 24 h [26].

Understanding whether the biofilm-forming capacity of MRSA strains could be positively or negatively correlated with any toxin genes is essential for the clinicians to better evaluate and manage this type of infection. There are a lack of data regarding this type of correlation. In this study, there was no significant correlation between the biofilm production and any kind of toxin genes, suggesting that the carriage of toxin genes may be an unreliable marker to be correlated with biofilm production. In contrast with our results, a recent study conducted in China indicated that virulent strains could be more likely to be strong biofilm producers [27].

The regulation of pathogenesis and biofilm development in MRSA strains is related to the *agr*-mediated quorum sensing system [28]. It was confirmed that there was a correlation between the *agr* activity and the biofilm formation, as previously documented [29]. Concerning the type of *agr* alleles, our results indicated that *agr* I in opposition to *agr* II and IV correlated negatively with the biofilm production. Several reports observed higher prevalence rates of *agr* II gene; that is, 63.9% in USA [30] and 44% in South Korea [31] among biofilm producing MRSA strains. However, this is not in agreement with previous studies in Belgium, where the biofilm formation in *S. aureus* isolated from bovine mastitis with *agr* I *type* was higher than those with other *agr* types [32,33]. The differences in this correlation might be attributed to the heterogeneity in the origins of MRSA strains. These observations support that there is no absolute correlation between *agr* alleles and biofilm production.

Biofilm associated infections are often associated with poor clinical responses and frequent relapses despite using different antibiotic therapies [34]. The strong resistance of microbial cells in biofilms demands new strategies to fight such infections and restore the efficacies of antibiotics. Recently, phages, QSIs, or physical methods have been proposed to eradicate the biofilm producing bacteria [35]. Our manuscript focused on the development of promising antibiofilm agents through inhibition of QS such as Zno NPs, proteinase K, and HAM. One of the antibiofilm strategies is to inhibit the attachment of the biofilm to motile surfaces using ZnO NPs, as they have catalytic, semi conducting, magnetic, and antimicrobial characters [36]. In the present study, a notable mean inhibitory capacity of ZnO NPs (99.73%) was observed. According to Mahamuni et al., ZnO nanoparticles exhibited significant percentages of *S. aureus* biofilm inhibition, proving its usage as a potential antibiofilm agent in the biomedical application [37].

In an earlier report, it was found that proteinase K can emulate the naturally produced protease and it can be used to enhance the biofilm dispersal through the cleavage of surface proteins [38]. Proteinase K has a pronounced effect as an antibiofilm material on dropping the biofilm capacity among the tested MRSA isolates, with a mean inhibitory capacity percentage of 99.41%. This is correlated with another study carried out in India, where biofilm inhibition capacity was 84% after using proteinase K [38].

The present study, for the first time, investigated that ZnO NPs and proteinase K induced remarkable down regulations in the expressions of both *ica*A and *agr* alleles among biofilm associated MRSA isolates. Only a recent study carried out in Iran documented that the sub-minimum inhibitory concentration of ZnO-Ag NPs significantly reduced the *ica*A gene expression in *S. aureus* strains [39]. Another study conducted in Korea evaluated the inhibitory effect of ZnO NPs on *Pesudomonase aeruginosa* biofilms genotypically [40], where the transcriptional and mutant data indicated that ZnO NPs inhibited *Pesudomonase aeruginosa* biofilm formation via the *czcRS* two-component system, which controls the QS-controlled phenotypes such as biofilm formation. This confirms the efficacy of ZnO NPs as an antibiofilm substance among a broad range of Gram positive and Gram negative bacterial isolates. In conjunction with our data presented here, that proteinase K down regulated the expressions of *ica*A and *agr* alleles, an earlier study also reported that proteinase K inhibited biofilm formation and dispersed the established biofilms via the *agr*-mediated detachment, and these actions may be attributed to the cleavage of surface structures [41]. This highlighted the role of proteinase K as an efficient biofilm removal agent that hinders *S. aureus* biofilm formation.

One of the alternative approaches is targeting the bacterial QS system, which allows cell-to-cell communication between bacteria [11]. So, the current study evaluated the role of HAM as a QSI substance on inhibiting the biofilm formation. The results revealed that HAM exerted a pronounced antibiofilm activity against biofilm associated MRSA isolates with a mean reducing capacity of 99.66%. Moreover, it exhibited a down regulation in the expression of *agr* gene with a mean value of fold change up to 0.22. This is similar to the findings reported in Belgium, where 90% of biofilm-producing MRSA were inhibited after using HAM [42]. In another study carried out in Belgium, expressions of *agrA* and *agrC* were only affected when HAM was used, suggesting that HAM specifically affected *S. aureus* biofilm susceptibility through blocking the TraP QS system [43]. This may be related to the fact that HAM serves as a QSI by inhibiting RAP phosphorylation. Blocking QS not only decreases the virulence, but also prevents biofilm formation [11].

In our current research, Zno NPs showed a strong antimicrobial activity on biofilm producing MRSA strains; however, no antimicrobial activities were recorded for proteinase K and HAM. The inhibitory effect of ZnO NPs proved in our study against biofilm producing MRSA isolates is similar to a previous study reported in India [44], where ZnO NPs revealed wide inhibition zone diameters against strong biofilm producing *S. aureus* isolates. The antibacterial activity of ZnO NPs is due to the reaction of ZnO surface with water, resulting in the production of hydroxyl radicals, which induce oxidative stress causing bacterial cell damage [44]. Therefore, Zno NPs can be used alone in the treatment of biofilm-mediated infections, but proteinase K or HAM may be co-administrated with other antimicrobials in fighting biofilm producing MRSA.

## 4. Materials and Methods

### 4.1. Ethical Statements

All human MRSA isolates used in this study were originally isolated from patient samples for the purpose of diagnosis and proper treatment. The samples were collected with a sole aim of patients care. Therefore, the ethical approval of participants was not necessary as all clinical and laboratory data were anonymized.

### 4.2. Isolation and Identification of MRSA

Five hundred samples were collected from animal (200), food (100), and human (200) sources from various localities in Sharkia province, Egypt. The animal sample types included milk from mastitic cows (*n* = 150) and fresh meat (*n* = 50), while the food samples comprised minced meat (*n* = 50), sausage (*n* = 25), and lunchun (*n* = 25). The human sample compositions were as follows: sputum (*n* = 53), pus (*n* = 47), urine (*n* = 40), blood (*n* = 28), CSF *(n* = 22, and peritoneal fluid (*n* = 10). All samples were subjected to microbiological examination for isolation of MRSA. All MRSA isolates were confirmed phenotypically based on standard bacteriological methods and susceptibility to methicillin and cefoxitin antibiotics [45] and genotypically depending on PCR analyses of 16S rRNA, *nuc*, and *mec*A genes [2,46].

### 4.3. Detection of Biofilm Producing MRSA

The in vitro biofilm production was measured phenotypically based on Congo red agar method [47] and adherence assay on a standard MTP [48] and genotypically depending on the detection of *ica*A gene [49].

### 4.4. Antimicrobial Susceptibility Testing of MRSA

The Kirby–Bauer disk diffusion method was used to detect the antimicrobial susceptibility of all MRSA strains. Thirteen types of antimicrobials discs (Oxoid, UK), including methicillin (ME; 5 μg), cefoxitin (FOX; 30 μg), vancomycin (VA; 30 μg), clindamycin (DA; 2 μg), rifamycin SV (RF; 30 μg), trimethoprim-sulfamethoxazole (SXT; 1.25/23.75 μg), imipenem (IPM; 10 μg), erythromycin (E; 15 μg), ciprofloxacin (CIP; 5 μg), ceftriaxone (CRO; 30 μg), amoxicillin-clavulanic acid (AMC; 20/10 μg), gentamicin (CN; 10 μg), and chloramphenicol (C; 30 μg), were used and the susceptibility of the tested strains was determined according to the guidelines of Clinical and Laboratory Standards Institute (CLSI) [50]. The MDR was defined as resistance to three or more different classes of the evaluated antimicrobials [51]. The minimum inhibitory concentration (MIC) of vancomycin was determined phenotypically using the broth micro dilution method [52].

### 4.5. Toxin Genes Profiles and agr Alleles of MRSA Isolates

Toxin gene profiles of all MRSA isolates were identified by detecting *sea-seg*, *eta*, *etb*, *tst*, and *pvl* genes using different multiplex PCR conditions [53]. Furthermore, *agr* alleles of the examined isolates were accomplished using a previously described multiplex PCR assay [54]. For the quality control of the genotyping methods, *E. coli* ATCC25922 and *S. aureus* ATCC25923 were included with all PCR runs as negative and positive controls, respectively.

### 4.6. Assessment of Antibiofilm and Antimicrobial Activities of Promising Agents

Promising antibiofilm agents including ZnO NPs, HAM, and proteinase K were purchased from Sigma-Aldrich (St. Louis, MO, USA). For assessing the ability of ZnO NPs to inhibit the biofilm formation of the tested isolates, wells of a microtitre plate were filled with 200 µL of the bacterial suspension in tryptone soya broth (TSB, Oxoid, UK), supplemented with 0.5% glucose and 5 mM ZnO NPs [39], and the plates were then incubated aerobically at 37 °C for 24 h. To evaluate the dispersion actions of HAM and proteinase K, the bacteria were incubated in sterile polystyrene 96-well plates with TSB medium containing 0.5% glucose at 37 °C for 24 h. After the removal of planktonic cells, 250 μM of HAM [55] and 2 μg/mL of proteinase K [38] were separately added to each well and the plates were incubated for another 24 h at 37 °C. Further, the protocol of Stepanovic et al. was then followed [49]. Three replicates were made for each treatment. All isolates were then analyzed for the quantitative production of biofilm using an enzyme-linked immunosorbent assay (ELISA) reader at 570 nm as described earlier [56] and for the expressions of *ica*A and the detected *agr* allele using qRT-PCR assay. Briefly, after inoculation of the tested isolates on TSB medium containing 0.5% glucose with or without the promising antibiofilm agents, cell pellets obtained after centrifugation were collected and used immediately for total RNA extraction using the RNeasy Mini Kit (QIAGEN Hilden, Germany) according to the manufacturer’s instructions. The bacterial aliquots without treatment were served as negative controls. The RNA quality was determined by agarose gel electrophoresis and RNA concentrations were estimated by measuring the absorbance at 260 and 280 nm using a NanoDrop ND-1000 spectrophotometer (Thermo Fisher Scientific Inc, Waltham, MA, USA). Purified RNA was immediately reverse transcribed to complementary DNA (cDNA) using the QuantiTect SYBR Green RT-PCR Kit (Qiagen GmbH, Hilden, Germany) according to the manufacturer’s recommended protocol. The qPCR was performed to quantify the fold changes in the expression of *ica*A and *agr* genes utilizing the QuantiTect SYBR Green PCR master mix (Qiagen GmbH, Hilden, Germany) in a Stratagene MX3005P real time PCR machine (Stratagene, La Jolla, CA, USA) under the specific thermal cycling conditions [50,54]. After the qPCR runs, we performed a final melting curve analysis to verify that the expected targets were actually amplified. Data were normalized to *16S rRNA* reference gene (an endogenous control) with the following primer sequence: Forward (5′-CGTGGAGGGTCATTGGA-3′) and Reverse (5′-CGTTTACGGCGTGGACT-3′) [57]. The data were then subjected to analysis using the Stratagene MX3005P software. The relative quantification of target gene expression levels was calculated using the 2^−ΔΔCt^ method [58]. Finally, the antimicrobial activities of the used agents were detected by the agar well diffusion technique [59].

### 4.7. Statistical Analyses

Correlation analyses were measured on the raw data after its conversion to a binary outcome (1 = variable presence, 0 = variable absence). Significance of correlation was analyzed at a significance level of 0.05. The variables that were identical among the isolates were excluded from the correlation analyses. The correlation analyses and visualization were estimated using R packages, *corrplot*, *heatmaply*, *hmisc*, and *ggpubr* [60,61,62].

## 5. Conclusions

From our results, a strong positive correlation was found between the biofilm production and antimicrobial resistances. Fortunately, promising antibiofilm activities were recorded for proteinase K or HAM, which can be co-administrated with other antimicrobials to reduce the risk of biofilm associated infections. In another context, we recommended the use of Zno NPs to reduce the risk of hospital infections, especially those occurring through the implantable devices such as catheters, because of the strong antimicrobial/antibiofilm activities related to Zno NPs on the biofilm producing MRSA strains.

## Figures and Tables

**Figure 1 antibiotics-09-00667-f001:**
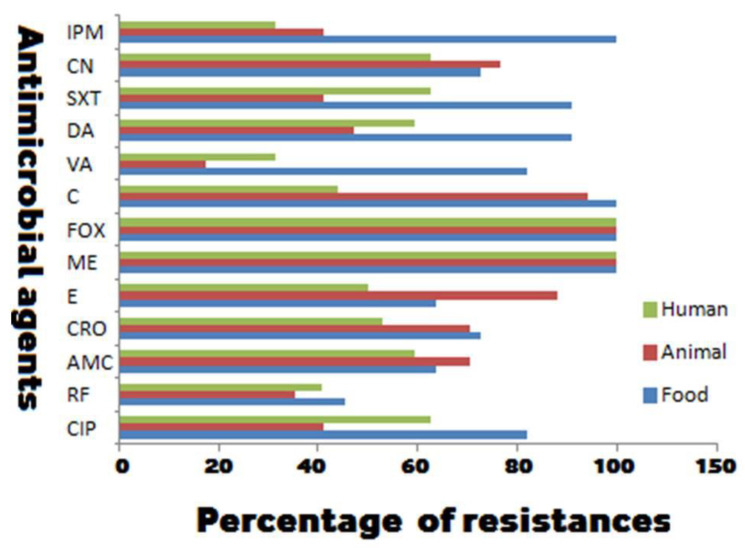
Antimicrobial resistance of biofilm producing methicillin-resistant *Staphylococcus aureus* isolates from different sources. DA: clindamycin; CN: gentamicin; VA: vancomycin; FOX: cefoxitin; CRO: ceftriaxone; IPM: imipenem; ME: methicillin; AMC: amoxicillin-clavulanic acid; RF: rifamycin SV; C: chloramphenicol; E: erythromycin; SXT: trimethoprim-sulfamethoxazole; CIP: ciprofloxacin.

**Figure 2 antibiotics-09-00667-f002:**
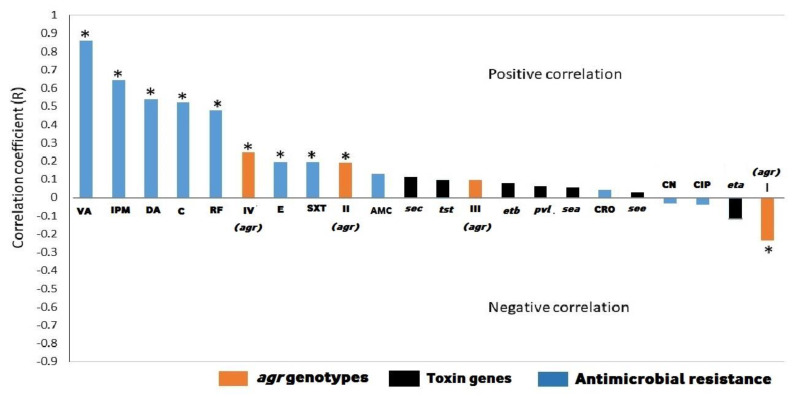
Correlation between biofilm production and antimicrobial resistance, toxin genes, and *agr* genotypes of methicillin-resistant *Staphylococcus aureus* isolates. VA: vancomycin; IPM: imipenem; DA: clindamycin; C: chloramphenicol; RF: rifamycin SV; E: erythromycin; SXT: trimethoprim-sulfamethoxazole; AMC: amoxicillin-clavulanic acid; CRO: ceftriaxone; CN: gentamicin; CIP: ciprofloxacin; *agr*: accessory gene regulator; *pvl*: Panton-Valentine leucocidin; *sea, see* and *sec*: staphylococcal enterotoxins A, B and C; *tst*: toxic shock syndrome toxin; *eta* and *etb*: exfoliative toxins A and B; ***** above the columns indicate significant differences (*p* < 0.05).

**Figure 3 antibiotics-09-00667-f003:**
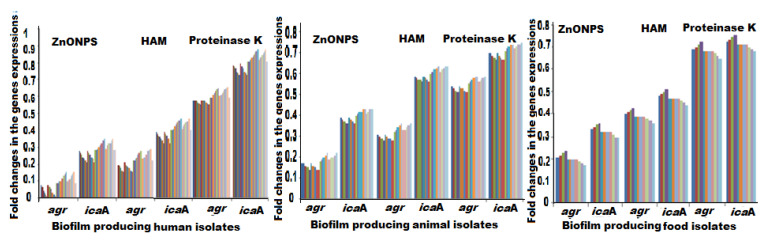
Fold changes in the expression of *ica*A and *agr* genes for all biofilm producing isolates post exposure to ZnO NPs, HAM, and proteinase K. *agr*: accessory gene regulator; *ica*A: intercellular adhesion A gene; ZnO NPs: zinc oxide nanoparticles; HAM: hamamelitannin.

**Table 1 antibiotics-09-00667-t001:** Detailed characteristics of biofilm and non-biofilm producing methicillin-resistant *Staphylococcus aureus* isolates from different sources regarding antimicrobial resistance and *agr* and toxin genes.

Antimicrobial Resistance/*agr* and Toxin Genes		No. (%) of MRSA Stains from Various Sources with Different Biofilm Phenotypes							
		Non-Biofilm Producers				Biofilm Producers			
		Human (84) *	Animal (49) *	Food (29) *	Total (162) *	Human (32) *	Animal (17) *	Food (11) *	Total (60) *
**Antimicrobial resistance ****	**CIP**	46 (54.8)	20 (41)	13 (44.8)	79 (48.8)	20 (62.5)	7 (41.2)	9 (81.8)	36 (60)
	**RF**	15 (17.9)	9 (18.4)	7 (24.1)	31 (19.1)	13 (40.6)	6 (35.3)	5 (45.5)	24 (40)
	**AMC**	38 (45.2)	22 (44.9)	17 (58.6)	77 (47.5)	19 (59.4)	12 (70.6)	7 (63.6)	38 (63.3)
	**CRO**	38 (45.2)	22 (44.9)	17 (58.6)	77 (47.5)	17 (53.1)	12 (70.6)	8 (72.7)	37 (61.7)
	**E**	33 (39.3)	19 (38.8)	16 (55.2)	68 (42)	16 (50)	15 (88.2)	7 (63.6)	38 (63.3)
	**C**	9 (10.7)	5 (10.2)	3 (10.3)	17 (10.5)	14 (43.8)	16 (94.1)	11 (100)	41 (68.3)
	**VA**	9 (10.7)	2 (4.1)	2 (6.9)	13 (8)	10 (31.3)	3 (17.6)	9 (81.8)	22 (36.7)
	**DA**	33 (39.3)	23 (46.9)	9 (31)	65 (40.1)	19 (59.4)	8 (47.1)	10 (90.9)	37 (61.7)
	**SXT**	39 (46.4)	18 (36.7)	11 (37.9)	68 (42)	20 (62.5)	7 (41.2)	10 (90.9)	37 (61.7)
	**CN**	40 (47.6)	19 (38.8)	10 (34.5)	69 (42.6)	20 (62.5)	13 (76.5)	8 (72.7)	41 (68.3)
	**IPM**	5 (6)	1 (2)	0 (0)	6 (3.7)	10 (31.3)	7 (41.2)	11 (100)	28 (46.7)
***agr* alleles**	***agr* I**	74 (88.1)	34 (69.4)	18 (62.1)	126 (77.8)	22 (68.8)	8 (47.1)	9 (81.8)	39 (65)
	***agr* II**	0 (0)	4 (8.2)	2 (6.9)	6 (3.7)	0 (0)	2 (11.8)	1 (9.1)	3 (5)
	***agr* III**	10 (11.9)	6 (12.2)	6 (20.7)	22 (13.6)	7 (21.9)	5 (29.4)	1 (9.1)	13 (21.7)
	***agr* IV**	0 (0)	5 (10.2)	3 (10.3)	8 (4.9)	0 (0)	2 (11.8)	3 (27.3)	5 (8.3)
**Toxin genes**	***pvl***	36 (42.9)	25 (51)	16 (55.2)	77 (47.5)	16 (50)	12 (70.6)	4 (36.4)	32 (53.3)
	***sea***	28 (33.3)	25 (51)	12 (41.4)	65 (40.1)	16 (50)	9 (52.9)	3 (27.3)	28 (46.7)
	***see***	11 (13.1)	5 (10.2)	3 (10.3)	19 (11.7)	4 (12.5)	3 (17.6)	1 (9.1)	8 (13.3)
	***sec***	10 (11.9)	1 (2)	1 (3.4)	12 (7.4)	4 (12.5)	0 (0)	4 (36.4)	8 (13.3)
	***tst***	7 (8.3)	5 (10.2)	3 (10.3)	15 (9.3)	6 (18.8)	3 (17.6)	1 (9.1)	10 (16.7)
	***eta***	10 (11.9)	9 (18.4)	7 (24.1)	26 (16)	2 (6.3)	2 (11.8)	0 (0)	4 (6.7)
	***etb***	5 (6)	3 (6.1)	5 (17.2)	13 (8)	4 (12.5)	3 (17.6)	1 (9.1)	8 (13.3)

MRSA: methicillin-resistant *Staphylococcus aureus*; CIP: ciprofloxacin; RF: rifamycin SV; AMC: amoxicillin-clavulanic acid; CRO: ceftriaxone; E: erythromycin; C: chloramphenicol; VA: vancomycin; DA: clindamycin; SXT: trimethoprim-sulfamethoxazole; CN: gentamicin; IPM: imipenem; *agr*: accessory gene regulator; *pvl*: Panton–Valentine leucocidin; *sea, see,* and *sec*: staphylococcal enterotoxins A, B, and C; *tst*: toxic shock syndrome toxin; *eta* and *etb*: exfoliative toxins A and B; *: the number in parentheses represents the corresponding number of biofilm phenotype for MRSA isolates from different sources; **: all strains were resistant to methicillin and cefoxitin.

**Table 2 antibiotics-09-00667-t002:** Summary of antibiofilm activities of promising antibiofilm agents against biofilm producing methicillin-resistant *Staphylococcus aureus* isolates from various sources.

Antibiofilm Activities		Promising	MRSA Isolates Sources (No.)		
		Antibiofilm Agents	Human (32)	Animal (17)	Food (11)
**Reduction percentage of biofilm capacity *** **(Phenotypically, MTP method)**		**ZnO NPs**	99.79 ± 0.089	99.81 ± 0.088	99.58 ± 0.016
		**HAM**	99.58 ± 0.072	99.81 ± 0.007	99.58 ± 0.016
		**Proteinase K**	99.51 ± 0.065	99.28 ± 0.013	99.44 ± 0.004
**Fold changes of biofilm and QS genes expressions**	***ica*A**	**ZnO NPs**	0.28 ± 0.042	0.18 ± 0.027	0.15 ± 0.020
		**HAM**	0.40 ± 0.042	0.32 ± 0.027	0.45 ± 0.020
		**Proteinase K**	0.81 ± 0.045	0.55 ± 0.027	0.69 ± 0.020
	***agr***	**ZnO NPs**	0.08 ± 0.042	0.40 ± 0.025	0.48 ± 0.020
		**HAM**	0.22 ± 0.042	0.60 ± 0.027	0.59 ± 0.020
		**Proteinase K**	0.59 ± 0.032	0.71 ± 0.027	0.82 ± 0.020

MTP: microtiter plate; QS: quorum sensing; *ica*A: intercellular adhesion A gene; *agr*: accessory gene regulator; ZnO NPs: zinc oxide nanoparticles; HAM: hamamelitannin; MRSA: methicillin-resistant *Staphylococcus aureus*. All data represent the mean values ± standard deviation for all analyzed biofilm producing MRSA isolates.

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
