# Peer review of "Promising Antibiofilm Agents: Recent Breakthrough against Biofilm Producing Methicillin-Resistant Staphylococcus aureus"

_antibiotics, 2020, doi:10.3390/antibiotics9100667_

Round 1
Reviewer 1 Report
The manuscript by El-Hamid et al presents a well designed study of MRSA isolates from a variety of sources and merits publication. However, there is limited originality in the study but more importantly the current version does not provide the proper context for the results. The main focus of this study, links between biofilm formation and pathogenicity and inhibition of biofilm by Zn/Proteinase K/HAM have been extensively studied. The authors reference some related studies but do not provide a clear explanation in their discussion of how their results add, or not, to our understanding of MRSA. For example, they quote a study from Nepal where biofilm and MDR were linked (L244 P15) in MRSA which is well documented but no detailed comparison provided. Similarly, in the next paragraph the authors state 'few studies' indicate virulence and biofilm are linked but only reference one study. Also unclear if there any novelty in their Zn or proteinase results? Why is the Zn inhibitory activity notable (L286 P18)?
Minor corrections
L44 P2 Correct 'proteinas'
L92 P4 Change to 'capable of'
L101 P5 Correct 'appointed'
L107 P5 Provide references
L108 P5 Correct 'arguments'
P169 P10 Correct to genotype
L287 P17 Correct sentence or provide experimental evidence
Author Response
Dear Professor Doctor/ Editor-in-Chief of Antibiotics Journal
The manuscript ID: antibiotics-927323
Title: Promising antibiofilm agents: recent breakthrough against biofilm producing methicillin-resistant Staphylococcus aureus
Many thanks for all your efforts to improve our manuscript. We would like to thank the reviewers for their raised and thorough comments. The corrections requested by the reviewers have been done point by point as shown in the revision form. Hopefully, our revised manuscript meets the expectations of you and the reviewers and be considered for publication in Antibiotics Journal.
Response to the comments:
Reviewer 1:
Comments and Suggestions for Authors
The manuscript by El-Hamid et al presents a well-designed study of MRSA isolates from a variety of sources and merits publication.
Thanks for your comment.
However, there is limited originality in the study but more importantly the current version does not provide the proper context for the results.
Our manuscript focuses on the development of promising antibiofilm agents such as Zno NPs, proteinase K and HAM to manage the MRSA biofilms and we endeavored ascertain whether these approaches could effectively interfere with QS systems of the examined MRSA isolates.
From our point of view, the originality in this paper is that our present study, for the first time, investigated that ZnO NPs and proteinase K induced remarkable down regulation in the expressions of both icaA and agr alleles among biofilm associated MRSA isolates confirming their efficacy as antibiofilm and QS inhibitor substances. All these findings were illustrated and proved in the results section by detaling the levels of their suppression of the expressions of icaA and agr genes. Moreover, the antibiofilm activites of the used substances were confirmed by observing the decrease in the capacity of all biofilm producing MRSA isolates post exposure to all examined substances as was detailed in the results section by mentioning their efficacy rates in percentages.
The main focus of this study links between biofilm formation and pathogenicity and inhibition of biofilm by Zn/Proteinase K/HAM have been extensively studied.
Thanks for your comment.
The authors reference some related studies but do not provide a clear explanation in their discussion of how their results add, or not, to our understanding of MRSA. For example, they quote a study from Nepal where biofilm and MDR were linked (L244 P15) in MRSA which is well documented but no detailed comparison provided.
The detailed comparison with this study was provided in addition to illustrating the interpretation for this comparison and how our study added to the understanding of this important finding as well as the attribution for that.
Similarly, in the next paragraph the authors state 'few studies' indicate virulence and biofilm are linked but only reference one study.
It was corrected to a recent study conducted in China.
Also unclear if there any novelty in their Zn or proteinase results?
This was demonstrated in lines 296-298 (The present study, for the first time, investigated that ZnO NPs and proteinase K induced remarkable down regulations in the expressions of both icaA and agr alleles among biofilm associated MRSA isolates). We added the evidences for these findings as the previous papers evaluating the effects of zinc oxide or proteinase K on icaA and agr genes expression stated the effects of theses substances on Staphylococcus aureus biofilm formation or dispersion either combined with other substances or only on the expression of either icaA and agr genes not both. All these data were detailed in their appropriate position in the discussion section as was the reviewer addressed.
Why is the Zn inhibitory activity notable (L286 P18)?
Both zinc oxide and proteinase K had notable inhibitory activities on Staphylococcus aureus biofilm formation and dispersion. The ZnO NPs inhibitory activity was mentioned in the original version. Therefore, we added, in the revised manuscript, the papers demonstrating the inhibitory effects for proteinase K on Staphylococcus aureus biofilms either inhibiting biofilm formation or dispersing the established biofilms and also interpreted these findings. Moreover, we also interpreted the already stated ZnO NPs biofilm inhibitory activities.
Minor corrections
L44 P2 Correct 'proteinas'
Done
L92 P4 Change to 'capable of'
Done
L101 P5 Correct 'appointed'
Done
L107 P5 Provide references
Done
L108 P5 Correct 'arguments'
Done
P169 P10 Correct to genotype
Done
L287 P17 Correct sentence or provide experimental evidence
The sentence was replaced by another one demonstrating more the comparison and interpreted it.
Reviewer 2 Report
This paper focuses on the characterization of different methicillin-resistant Staphylococcus aureus (MRSA) bacterial colonies, isolates from different animal, food and human sources. In particular, the study aimed to isolate the colonies of biofilm-forming bacteria from the no-biofilm-forming ones, in order to underline their differences for how concern: i) their resistance against commonly employed biocides; ii) their agr and iii) toxigenes profiles. As expected, the biofilm forming MRSA, in general, demonstrated a higher resistance against the tested biocides. For this reason, three promising biofilm inhibitors (Zno NPs, proteinase K and HAM) have been tested against the MRSA biofilms, confirming their efficacy. Moreover, the biocidal action of these three compounds have been tested: only the Zno NPs demonstrated good biocidal activities, while for proteinase K and HAM no biocidal activities have been observed.
In general, I found the research interesting, especially for how concern the main results obtained (the biocides-resistance profiles of MRSA biofilm and no-biofilm producers, the testing of the biofilm inhibitors and the evaluation of their biocidal activity). Anyway, I have had difficulties in the interpretation of some results and data, more specifically, in the paragraphs “characterization of MRSA biofilm producers” and “Biofilm associated genes’ expressions post exposure to the antibiofilm agents”.
I think that the results are interestingly and appropriately described in the discussion part.
More specific comments are reported below:
Abstract: I have never seen an abstract divided in paragraphs. I think that will be more appropriated to delete and unify background, methods and results in a single paragraph.
Please, change the style of the document and name each paragraph with the corresponding number (for example: 1. Introduction, 2. Results, 2.1. Characterization of MRSA biofilm producers etc). I suggest using the template proposed by the Journal.
Introduction: The theme of the MRSA and the relation between the biofilm formation and the increasing of their resistance against antimicrobials is extensively explained. Anyway, I suggest describing more deeply the methods employed and the reasons why you decided to use ZnO NPs, proteinase K and HAM in the experimentation (I think that this part must be included also in the introduction, in order to facilitate the comprehension of the article, and not only in the discussion).
Line 71: replace the comma with a dot.
Line 116: in the paragraph Characterization of MRSA biofilm producers I cannot find a relation between the described results and the ones reported in Table 1. In line 120 is said that the results of Congo red agar identification can be found in Table 1. But this seems controversial, because in Table 1 it is possible to see only the classification of the colonies. Whit this statement, is it meant that the Congo red agar results are implicit in the Table 1? Moreover, I didn’t understand the percentage reported (for example human samples 16%) and where these results come from.
Line 133: “Regarding the biofilm producing MRSA, all these isolates were insusceptible to 3 or more antimicrobials from different classes suggesting them to be MDR.” I understood what it is meant. But I think that this statement can create some confusion: observing the Table 1, it seems that only in the case of biofilm forming MRSA obtained from food it is possible to find a correlation with what has been stated (100% in the case of C and IPM). I suggest changing the sentence and underlining that, for each group, at least one of the isolates demonstrated to be insusceptible to 3 or more antimicrobials.
Line 228: replace “well” with “gel”.
From line 212: I think that the results reported in the following paragraph must be written more clearly because I didn’t understand what it means, especially when you speak about the suppression of icaA gene and the downregulation of agr gene.
Author Response
Dear Professor Doctor/ Editor-in-Chief of Antibiotics Journal
The manuscript ID: antibiotics-927323
Title: Promising antibiofilm agents: recent breakthrough against biofilm producing methicillin-resistant Staphylococcus aureus
Many thanks for all your efforts to improve our manuscript. We would like to thank the reviewers for their raised and thorough comments. The corrections requested by the reviewers have been done point by point as shown in the revision form. Hopefully, our revised manuscript meets the expectations of you and the reviewers and be considered for publication in Antibiotics Journal.
Response to the comments:
Reviewer 2:
Comments and Suggestions for Authors
This paper focuses on the characterization of different methicillin-resistant Staphylococcus aureus (MRSA) bacterial colonies, isolates from different animal, food and human sources. In particular, the study aimed to isolate the colonies of biofilm-forming bacteria from the no-biofilm-forming ones, in order to underline their differences for how concern: i) their resistance against commonly employed biocides; ii) their agr and iii) toxigenes profiles. As expected, the biofilm forming MRSA, in general, demonstrated a higher resistance against the tested biocides. For this reason, three promising biofilm inhibitors (Zno NPs, proteinase K and HAM) have been tested against the MRSA biofilms, confirming their efficacy. Moreover, the biocidal action of these three compounds have been tested: only the Zno NPs demonstrated good biocidal activities, while for proteinase K and HAM no biocidal activities have been observed.
Thanks for your valuable and comprehensive presentation for the manuscript.
In general, I found the research interesting, especially for how concern the main results obtained (the biocides-resistance profiles of MRSA biofilm and no-biofilm producers, the testing of the biofilm inhibitors and the evaluation of their biocidal activity).
Thanks for your comment.
Anyway, I have had difficulties in the interpretation of some results and data, more specifically, in the paragraphs “characterization of MRSA biofilm producers” and “Biofilm associated genes’ expressions post exposure to the antibiofilm agents”.
All of them were taken into consideration in the revised manuscript and also were illustrated well in the response of the following comments.
I think that the results are interestingly and appropriately described in the discussion part.
Thanks for your comment.
More specific comments are reported below:
Abstract: I have never seen an abstract divided in paragraphs. I think that will be more appropriated to delete and unify background, methods and results in a single paragraph.
Done
Please, change the style of the document and name each paragraph with the corresponding number (for example: 1. Introduction, 2. Results, 2.1. Characterization of MRSA biofilm producers etc). I suggest using the template proposed by the Journal.
Done
Introduction: The theme of the MRSA and the relation between the biofilm formation and the increasing of their resistance against antimicrobials is extensively explained.
Thanks for your comment.
Anyway, I suggest describing more deeply the methods employed and the reasons why you decided to use ZnO NPs, proteinase K and HAM in the experimentation (I think that this part must be included also in the introduction, in order to facilitate the comprehension of the article, and not only in the discussion.
All the reviewer asked were detailed in the introduction section regarding the methods employed and the reasons why we decided to use ZnO NPs, proteinase K and HAM in the experimentation and accordingly, we improved the aim of the study.
Line 71: replace the comma with a dot.
Done
Line 116: in the paragraph Characterization of MRSA biofilm producers I cannot find a relation between the described results and the ones reported in Table 1. In line 120 is said that the results of Congo red agar identification can be found in Table 1. But this seems controversial, because in Table 1 it is possible to see only the classification of the colonies. Whit this statement, is it meant that the Congo red agar results are implicit in the Table 1?
Thanks for your comment. It is actually correct and the citation for table 1 in this position was removed.
Moreover, I didn’t understand the percentage reported (for example human samples 16%) and where these results come from.
This point was illustrated more clearly in this position to can know where these results come from.
Line 133: “Regarding the biofilm producing MRSA, all these isolates were insusceptible to 3 or more antimicrobials from different classes suggesting them to be MDR.” I understood what it is meant. But I think that this statement can create some confusion: observing the Table 1, it seems that only in the case of biofilm forming MRSA obtained from food it is possible to find a correlation with what has been stated (100% in the case of C and IPM). I suggest changing the sentence and underlining that, for each group, at least one of the isolates demonstrated to be insusceptible to 3 or more antimicrobials.
The sentence was already changed to (Regarding the biofilm producing MRSA, all these isolates were resistant to three or more different categories of the investigated antimicrobials suggesting them to be MDR) to avoid the conflict. In our study, we have investigated a total of 13 antimicrobials of various groups, so even if the susceptibility percentages for 2 antibiotics reached 100%, there were other 3 antimicrobials or more from different classes to which the isolates were resistant to can categorize them as MDR. Eventhough, we changed the sentence as the reviewer asked.
Line 228: replace “well” with “gel”.
From our point of view, in accordance with other scientific papers which documented that the agar well diffusion method is adopted in antimicrobial activity assay. For example: Magaldi et al. (2004) stated that the agar well diffusion method was widely used to evaluate the antimicrobial activity of plants or microbial extracts.
Magaldi S, Mata-Essayag S, Hartung de Capriles, C. et al. (2004): Well diffusion for antifungal susceptibility testing. Int. J. Infect. Dis., 8: 39-45.
Meanwhile, AGID (agar gel immunodiffusion assay) is a diagnostic test using serum (the fluid, non-cellular part of blood0 that detects antibody produced in response to infection (Linh et al., 2003).
Linh BK, Thuy DT, My LN, Sasaki O and Yoshihara S (2003): Application of Agar Gel Diffusion Test to the Diagnosis of Fasciolosis in Cattle and Buffaloes in the Red River Delta of Vietnam. Japan Agricultural Research Quarterly 37(3): 201-205.
Therefore, the well should be as it is and cannot be replaced by gel according to the use in this study.
From line 212: I think that the results reported in the following paragraph must be written more clearly because I didn’t understand what it means, especially when you speak about the suppression of icaA gene and the downregulation of agr gene.
It was re-written to be clearly, especially the suppression of icaA gene and the downregulation of agr gene.
Round 2
Reviewer 1 Report
The authors have adequately addressed most of the issues. However, since the novel of the study lies almost exclusively in the finding that 'ZnO NPs and proteinase K induced remarkable down regulation in the expressions of both icaA and agr alleles among biofilm associated MRSA isolates' the data supporting this results needs further elaboration. Full details of the results and methodology including controls need to be included, Table 2 does not give any indication of the qRT-PCR data variability or correlation analysis. This data and analysis could be added as supplementary figures/table.
Author Response
Dear Professor Doctor/ Editor-in-Chief of Antibiotics Journal
The manuscript ID: antibiotics-927323
Title: Promising antibiofilm agents: recent breakthrough against biofilm producing methicillin-resistant Staphylococcus aureus
On behalf of all the contributing authors, we would like to express our sincere appreciations of your letter and reviewers’ constructive comments concerning our article. These comments are all valuable and helpful for improving our article. According to the reviewers’ comments in the round 2, we have made extensive modifications to our manuscript and supplemented extra data to make our results convincing. In this revised version, changes to our manuscript were all green highlighted within the document. Point-by-point responses to the two nice reviewers are listed below this letter.
Response to the comments:
Reviewer 1:
Comments and Suggestions for Authors
The authors have adequately addressed most of the issues.
Thanks for your comment. We feel great thanks for your professional review work on our article.
However, since the novel of the study lies almost exclusively in the finding that 'ZnO NPs and proteinase K induced remarkable down regulation in the expressions of both icaA and agr alleles among biofilm associated MRSA isolates' the data supporting this results needs further elaboration. Full details of the results and methodology including controls need to be included, Table 2 does not give any indication of the qRT-PCR data variability or correlation analysis. This data and analysis could be added as supplementary figures/table.
Thank you again for your positive comments and valuable suggestions to improve the quality of our manuscript. We have responded specifically to each suggestion. Firstly, we detailed the methodology for qRT-PCR. Moreover, we made a correlation analysis between the expression of both agr and icaA genes among all isoaltes from different sources because the agr QS signaling system regulates the biofilm formation in S. aureus. Finally, our analysis led to supplementation of another figure to our results named Figure 3 to elaborate the expressions of both icaA and agr alleles among biofilm associated MRSA isolates from all sources and all its data were interpreted also in the results section.
Reviewer 2 Report
Thank you very much for modifying the manuscript in relation to the proposed comments.
I apologize for the comment concerning the agar well diffusion method and thank you for the clarifications.
I think that the manuscript can be accepted in the present form, except for a little mistake concerning the style at lines 122 and 131, where the number 2 should be associated to the introduction paragraph.
Author Response
Dear Professor Doctor/ Editor-in-Chief of Antibiotics Journal
The manuscript ID: antibiotics-927323
Title: Promising antibiofilm agents: recent breakthrough against biofilm producing methicillin-resistant Staphylococcus aureus
On behalf of all the contributing authors, we would like to express our sincere appreciations of your letter and reviewers’ constructive comments concerning our article. These comments are all valuable and helpful for improving our article. According to the reviewers’ comments in the round 2, we have made extensive modifications to our manuscript and supplemented extra data to make our results convincing. In this revised version, changes to our manuscript were all green highlighted within the document. Point-by-point responses to the two nice reviewers are listed below this letter.
Response to the comments:
Reviewer 2:
Comments and Suggestions for Authors
Thank you very much for modifying the manuscript in relation to the proposed comments.
We feel great thanks for your professional review work on our article.
I apologize for the comment concerning the agar well diffusion method and thank you for the clarifications.
Thank you for your kind comment.
I think that the manuscript can be accepted in the present form, except for a little mistake concerning the style at lines 122 and 131, where the number 2 should be associated to the introduction paragraph.
Thanks for your nice suggestions. We have formatted all these issues.